# Differential PARP inhibitor responses in *BRCA1*-deficient and resistant cells in competitive co-culture

Shiella A. Soetomo[1,2‡], Michael F. Sharp[1,2‡], Wayne Crismani [1,2*]

1 St Vincent's Institute of Medical Research, Fitzroy, Victoria, Australia, 2 The Faculty of Medicine, Dentistry and Health Sciences, The University of Melbourne, Victoria, Australia

‡ Shiella Soetomo and Michael Sharp should be recognised as co-first authors.
* wcrismani@svi.edu.au

## Abstract

Synthetic lethality describes a genetic relationship where the loss of two genes results in cell death, but the loss of one of those genes does not. Drugs used for precision oncology can exploit synthetic lethal relationships; the best described are PARP inhibitors which preferentially kill *BRCA1*-deficient tumours preferentially over *BRCA1*-proficient cells. New synthetic lethal targets are often discovered using genetic screens, such as CRISPR knockout screens. Here, we present a competitive co-culture assay that can be used to analyse drugs or gene knockouts with synthetic lethal effects. We generated new *BRCA1* isogenic cell line pairs from both a triple-negative breast cancer cell line (SUM149) and adapted pre-existing non-cancerous *BRCA1* isogenic pair (RPE). Each cell line of the isogenic pair was transformed with its own fluorescent reporter. The two-coloured cell lines of the isogenic pair were then grown together in the same vessel to create a more competitive environment compared to when grown separately. We used four PARP inhibitors to validate the ability to detect synthetic lethality in *BRCA1*-deficient cancer cells. The readout of the assay was performed by counting the fluorescently coloured cells after drug treatment using flow cytometry. We observed preferential targeting of *BRCA1*-deficient cells, by PARPi, at relative concentrations that broadly reflect clinical dosing. Further we reveal subtle differences between PARPi resistant lines compared to *BRCA1*-proficient cells. Here, we demonstrate the validation and potential use of the competitive assay, which could be extended to validating novel genetic relationships and adapted for live cell imaging.

## Introduction

Cancer cells can have genetic driver mutations that can make them heavily reliant on certain pathways for survival [1]. By targeting a second complementary gene or pathway that is synthetic lethal with the cancer-specific mutation, specific inhibitors can

**Data availability statement:** The data is attached to this revised submission.

**Funding:** W.C. receives a fellowship related to this work from the Victorian Cancer Agency (MCRF21006). SVI receives Operational Infrastructure Support from the Victorian State Government. S.S. received an Indonesian Lembaga Pengelola Dana Pendidikan scholarship. The funders had no role in study design, data collection and analysis, decision to publish, or preparation of the manuscript.

**Competing interests:** No.

selectively kill cancer cells while sparing healthy cells lacking the mutation [2]. This has been exemplified clinically with poly ADP-ribose polymerase inhibitors (PARPi) used to treat cancers with mutations in genes required for homologous recombination – most notably *BRCA1* and *BRCA2* [3]. While the development of PARPi have been breakthrough treatments for precision oncology, the field is expanding to discover inhibitors that target genetic vulnerabilities in cancers in pathways such as DNA repair, the cell cycle, epigenetic regulation, metabolic dependencies and oncogene-specific vulnerabilities [4–8]. Many of the synthetic lethal treatments currently under clinical investigation were discovered through genetic screens in cancer cells, for example using CRISPR-based screens [9]. Once a candidate gene is identified, it can be validated with an isogenic cell line pair [10].

Previous studies have demonstrated synthetic lethality using competitive growth assays with isogenic cells. In one approach, a CRISPR knockout was performed for one gene in a background where its synthetic lethal partner was already absent. A fluorescent reporter in the CRISPR construct allows tracking the relative fitness of the knockout cells when mixed in the pooled cell population [10]. Competitive growth assays can incorporate additional layers of complexity that can more closely represent *in vivo* environments by growing fluorescently labelled isogenic cells in 3D culture. Freischel et al. [11] observed how different cell types can out-compete each other based on their metabolic activity in certain growth conditions. The cells were grown in 3D spherical cultures that were observed periodically which enable the development of a mathematical model describing cell growth rates of the competing cells.

Previous synthetic lethal competitive growth assays using *BRCA1*-deficient cells have been used to validate new synthetic lethal relationships and PARPi resistance mechanisms [12–14]. Competitive growth assays have also been used to demonstrate synthetic lethality with other gene pairs – for example, *BRCA2* and *APEX* [15] – and may be useful to identify additional pairs and testing targeted drugs designed to exploit these relationships.

In the present study, we developed a two-colour competitive growth system that can be used to screen synthetic lethal inhibitors. We used four well-characterised PARPi and two pairs of isogenic cell lines to validate the assay. The assay demonstrated the expected synthetic lethal effect of the PARPi on *BRCA1* deficient cells, suggesting that this assay could be valuable in the future efforts to identify synthetic lethal inhibitors.

## Methods

### Cell culture

HEK293T cells were obtained from Andrew Deans (St. Vincent's Institute of Medical Research), SUM149PT, SUM149.B1.S*, SUM149 *53 BP1*[-/-] and SUM149 *SHLD1*[-/-] cells were generously provided by Stephen Pettitt (The Institute of Cancer Research) and the RPE *p53*[-/-] *BRCA1*[-/-] isogenic pair of cells were obtained from Alan D'Andrea (Dana Faber Cancer Center). HEK293T cells were cultured in high-glucose-supplemented Dulbecco's modified Eagle's medium (DMEM) (Sigma-Aldrich, St. Louis), supplemented with 10% (v/v) fetal bovine serum (FBS). SUM149 cell lines

were cultured in Ham's F-12 nutrient mix (Gibco, ThermoFisher Scientific, Waltham) supplemented with 10% (v/v) FBS, 1 $\mu$g/mL hydrocortisone, and 5 $\mu$g/mL Insulin. RPE cells were cultured in DMEM/Ham's nutrient mixture F-12 (Sigma) supplemented with 10% FBS. All cell lines were cultured in a humidified incubator at 37˚C in the presence of 5% $CO_2$. Cell doubling times were estimated using Incucyte° live cell imager.

### PARP inhibitors

Olaparib, niraparib, veliparib and talazoparib were all purchased from Selleck Chemicals. Inhibitor stock solutions were made in DMSO at an inhibitor concentration of 10 µM. Compound stocks were aliquoted and stored at −20˚C until further use.

### Isogenic cell lines

SUM149.A22.$BRCA1^{+/-}$, an isogenic partner of SUM149.PT.$BRCA1^{-/-}$, was generated at Monash Genome Modification Platform (MGMP) by reinserting a thymine corresponding to the missing nucleotide $BRCA1$ c.2169delT. Further, silent BseYI and AvaI restriction sites were inserted to facilitate genotyping of the editing cell line. The profile of the isogenic cell line was confirmed by PCR using oligonucleotide primer pair 5´-CAACTCATGGAAGGTAAAGAACC and 5´-AAAGCCTTCTGTGTCATTTCT, then followed by a restriction digest with AvaI. Further characterization was done via a sulforhodamine B cytotoxicity assay to show expected resistance to PARPi.

### Lentiviral transformation with fluorescent reporters

To produce lentivirus, HEK293T cells were seeded at a density of $4 \times 10^5$ cells in 1 mL suspension in a 6-well plate. After 24 h, a transfection mixture containing 375 ng of psPAX2, 125 ng of pVSV-G, and 500 ng of lentiviral vector was transfected into the cells (pLV-azurite, pLV-mCherry or pLV-eGFP) (respective addgene reference numbers; 36086, 36084 and 36083). The transfection particles were formed with 3 µL of FuGENE® HD in 100 µL of OptiMEM (Gibco, Thermo Fisher, USA) containing the DNA mixture from above. The transfection particles were left to form over 20 mins at room temperature. The transfection mixture was then added into the HEK293T cells. After 48 h and 72 h following transfection, the supernatant was harvested and pooled, then filtered through a 0.45 µm syringe filter, aliquoted and stored at −80˚C until further use.

Lentivirus transduction was performed by seeding the target cells at a density of $5 \times 10^4$ cells and immediately mixing with 300 µL of fluorescent protein encoding lentivirus with polybrene at a concentration of 8 µg/mL. After 24 h, cells were sorted using FACS Aria Fusion flow cytometer to gain a population of cells expressing the fluorescent transgene. These were the cells to be used in the competitive growth assay. eGFP containing virus was transduced into SUM149. A22 $BRCA1^{+/-}$ and RPE $p53^{-/-}$ $BRCA1^{+/+}$ cells. mCherry containing virus was transduced into SUM149PT $BRCA1^{-/-}$ and RPE $p53^{-/-}$ $BRCA1^{-/-}$. Azurite was transduced into SUM149 B1.s* $BRCA1^{D80bp/-}$, SUM149 $BRCA1^{-/-}$ $53 Bp1^{-/-}$ and SUM149 $BRCA1^{-/-}$ $SHLD1^{-/-}$.

### Sulforhodamine B assay

A drug-response sulforhodamine B (SRB) assay was performed in a 96-well plate. Both SUM149 and RPE cells were seeded at 1,500 cells per well with a total volume of 100 µL. After 24 h, the cells were dosed with a PARPi. The concentration range was 0.2 µM to 50.0 µM containing 6 concentrations with a 1/3 dilution factor. Following a 72 h incubation period at 37˚C with 5% $CO_2$, cells were again exposed to the same concentration of drug. The assay was stopped for SRB staining 48 h after the second dose. The media in each well was discarded and cells were fixed with 100 µL of 10% Trichloro-acetic Acid (TCA). Fixing of the cells took place for 1 h at 4˚C. The plate was washed with water three times and dried at room temperature. Once the plates were dry, 100 µL of 0.4% SRB in 0.1% acetic acid was added into each well and the

plate was incubated at room temperature for 30 mins. The SRB solution was then decanted. The plate was washed twice with 1% acetic acid and dried at room temperature. Once dried, 100 µL of 10 mM Tris-HCl pH 7.5 was added into each well and the plate was incubated for 5 minutes at room temperature. The absorbance was then read using EnSpire plate reader (Revvity) at 550 nm. The data was normalized by calculating the normalized percentage of survival.

$$\text{Normalized percentage of survival} = \left(1 - \left(\frac{\text{Abs } i - \text{Avg of Control Abs}}{\text{Avg of Blank Abs} - \text{Avg of Control Abs}}\right)\right) \times 100$$

Data plotting and analysis was then performed in GraphPad Prism. A semi log curve was fitted to the data sets and IC50 values were calculated. Area under the curve was calculated from the xy scatter plots in GraphPad Prism.

### Competitive growth assay

The pair of fluorescently labelled isogenic cells were seeded at 1:1 ratio in a 96 well plate at a total cell density of 1500 cells per plate. The cells were dosed with PARPi one day and four days after cell seeding. Six days after seeding, the cells were washed with Dulbecco's PBS, then trypsinised with 0.25% Trypsin-EDTA solution and resuspended in fresh media. eGFP, azurite and/or mCherry positive cells were then quantified using BD Fortessa HTS flow cytometer. The data was normalised by calculating the normalized percentage of survival.

$$\text{Normalised percentage of survival} = \left(1 - \left(\frac{\text{Cell count} - \text{Avg of untreated control}}{\text{Avg of no cell control} - \text{Avg of untreated control}}\right)\right) \times 100$$

### Western blotting

A confluent T25 flask was harvested of each cell line lysed by sonication in RIPA buffer. 40 µg of protein lysate was loaded onto 3–8% Tris Acetate NuPage™ polyacrylamide gel (Invitrogen). Gel was run for 60 mins at 150 V in Tris Acetate running buffer (Invitrogen). Proteins were transferred from the gel to Immobilon® FL PVDF membrane (Merck) using wet tank transfer (Bio-Rad) overnight at 30 V in 25 mM Tris, 192 mM glycine and 10% methanol. Revert™ 700 total protein staining (Li-Cor) was performed on the membrane for normalisation. Membrane was then blocked in 0.5% skim milk in PBS. Membrane was incubated with anti-BRCA1 (22362–1-AP, proteintech) for overnight at 4°C, then washed 4x in PBS. Membrane was then incubated for 60 mins at room temperature with anti-rabbit IgG DyLight™ 800 1:5000 (SA5–10036, Invitrogen) followed by a further 4 washes in PBS. Membrane was imaged using Li-Cor Odyssey CLx.

## Results

### Development of an isogenic *BRCA1*-proficient and *BRCA1*-deficient cell line

To develop the competitive growth assay, we first generated a gene edited SUM149PT *BRCA1*-/- cell line. The SUM149PT cell line was originally derived from a breast ductal carcinoma and contained a homozygous *BRCA1* pathogenic variant, c.2169delT, which resulted in a frameshift that led to a premature stop codon (p.P724Lfs*12). One allele at the *BRCA1* locus was corrected using CRISPR-Cas9 with a gRNA targeting near the pathogenic variant site and a homologous repair template. The homologous repair template contained the corrected *BRCA1* sequence and three silent variants to generate restriction enzymes sites for genotyping. The edited clone was named SUM149.A22 (S1 Fig). We tested SUM149.A22 for restoration of *BRCA1* function with an SRB cytotoxicity assay (Fig 1), in which we expected resistance to the PARPi, olaparib.

### Competitive growth assay development for assessment of synthetic lethality

We transduced both SUM149PT *BRCA1*-/- vs SUM149.A22 cells with lentivirus encoding mCherry and eGFP. We used four clinically relevant PARPi to validate our competitive growth assay. In parallel, we performed a SRB assay to compare

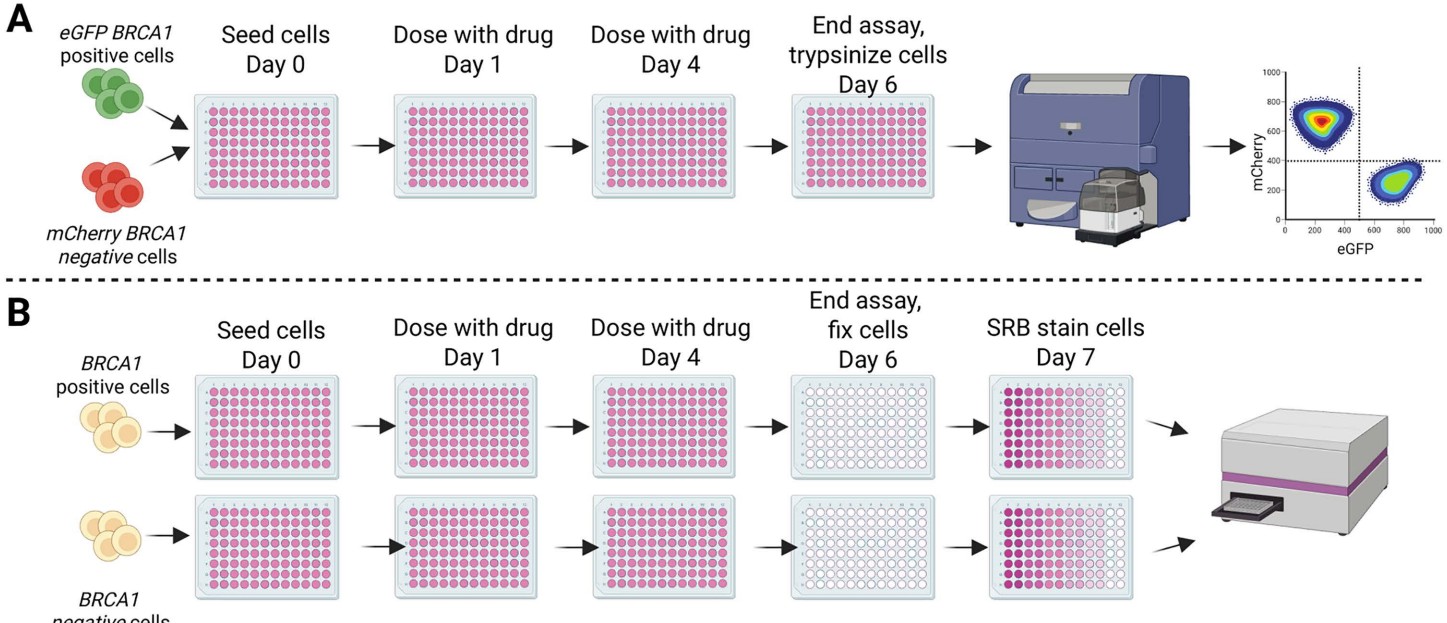

**Fig 1. Schematic of the competitive growth assay vs SRB assay. A)** The competitive growth assay procedure used fluorescently labelled isogenic cells grown together in a 96-well microplate. The readout out of the assay was by flow cytometry. **B)** The SRB assay procedure used isogenic cells grown in separate microplates. The assay was colorimetric and absorbance of stained cells was read on a microplate reader. Created in BioRender. Crismani, **W.** (2025) https://BioRender.com/h00p753.

the results between our new competitive growth assay and a traditional cytotoxicity assay. The schematics of each assay iare shown in Fig 1A and 1B, where the two key differences in the methods are: 1) the competitive growth assay involves both of the isogenic cell pair being grown in the same culture, 2) the endpoint of the competitive growth assay uses flow cytometry to count the eGFP SUM149.A22 and the mCherry SUM149 parental cells, whereas the SRB assay involves fixing and staining of the cells in a microplate and subsequent absorbance reading to estimate relative cell survival.

Both assays were performed as endpoint assays with the same 10-point olaparib dosage and concentration range. The competitive growth assay demonstrated clear synthetic lethality between the *BRCA1* isogenic SUM149 cells lines and is consistent with what was observed with the SRB assay (Fig 2A and 2B). The area under the curve (AUC) fold difference between the isogenic cell line pair for the competitive growth and SRB assay was 11.1 and 11.8, respectively. However, a key difference is that the IC50 values calculated from the SRB assay were much lower than the IC50 values from competitive growth assays. Further, the fold difference between IC50 values of each pair in the competitive growth assay was lower than the SRB assay with a fold change of 28.0 and 83.6, respectively. This demonstrates that the synthetic lethal window, which is the dosage range where olaparib kills the *BRCA1* negative cells but not the *BRCA1* positive cells, is larger in the SRB assay, when using IC50's as a measurement of potency. A noticeable difference between the survival curves in both assays, is observed between olaparib doses of 0.5 and 5 µM in the competitive growth assay, where the SUM149.A22 cells have a competitive advantage over SUM149.PT.*BRCA1$^{-/-}$* significantly increasing survival from 112.4% to 132.5% (p<0.0001). To further support that BRCA1 function has been restored in the SUM149.A22 cells, BRCA1 western blot was performed to show restored BRCA1 expression (S3 Fig). We therefore conclude based on the genotyping, western blot and the above functional data that the SUM149.A22 has a single copy of *BRCA1* restored and can be referred to as SUM149.A22 *BRCA1$^{+/-}$*.

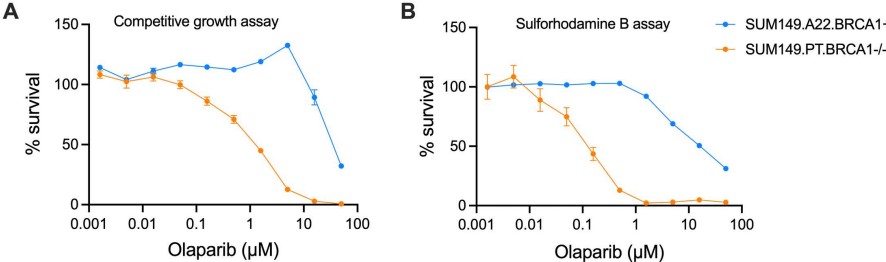

**Fig 2. Result comparison between the competitive growth assay and the SRB assay.** Comparison between the competitive growth assay (A) and sulforhodamine B assay (B) using the isogenic SUM149 *BRCA1* cell lines. The cells were dosed with olaparib twice over 6 days. The dosage range was between 0.0015 and 50 µM. Data is plotted as mean ± SEM where n = 8 for each treatment group. The IC50's in the competitive growth assay were 1.23 µM and 34.5 µM for the *BRCA1*$^{-/-}$ and *BRCA1*$^{+/-}$ cells, respectively. The IC50's in the SRB assay were 0.11 µM and 9.01 µM for the *BRCA1*$^{-/-}$ and *BRCA1*$^{+/-}$ cells, respectively. Area under the curve in the competitive growth assay was 351.8 and 3889.0 for the *BRCA1*$^{-/-}$ and *BRCA1*$^{+/-}$ cells, respectively. The area under the curve for SRB assay was 209.7 and 2475 for the *BRCA1*$^{+/-}$ and *BRCA1*$^{+/-}$ cells, respectively.

To further validate the competitive growth assay, we used two *BRCA1* isogenic cell line pairs; the *BRCA1*-deficient SUM149 breast cancer cell lines (as mentioned above) and a pair of retinal pigment epithelial cells (RPE) p53$^{-/-}$ with and without *BRCA1* knocked out. We tested the assay with four clinically relevant PARPi; olaparib, niraparib, veliparib and talazoparib. AUC was determined for each dose response due to some of the IC50 values being outside of the dose range used in the assay. To measure the strength of synthetic lethality with each drug, we calculated fold change between the AUC values in each isogenic pair.

The responses to the four PARPi in both isogenic pairs of SUM149 and RPE cell lines in the competitive growth assay demonstrated similar results (Fig 3A and 3B). Olaparib had the greatest measure of synthetic lethality in both the SUM149 and RPE isogenic pairs with an AUC fold change of 9.8 and 17.69, respectively. Consistent with observations in Fig 2A, *BRCA1*-proficient cells showed a growth advantage – evident as increased cell numbers – at low concentrations of olaparib and niraparib in the SUM149 background, and of olaparib, veliparib and niraparib in the RPE background. Both cell lines were most sensitive to talazoparib, where the *BRCA1* mutants had less than 16.6% survival at the lowest dose in used in the assay.

## PARP inhibitor resistance mechanism assessment with the competitive growth assay

The competitive growth assay can be used to assess specific mutations, or selected clones, that could lead to synthetic lethal resistance. We used a panel of PARPi-resistant cells in a SUM149 background developed in Dréan *et al.* [16] and Noordermeer *et al.* [14] to validate that PARPi resistance can be observed with the competitive growth assay. The SUM149.B1.s* has an 80 bp deletion which restores the open reading frame of *BRCA1* leading to BRCA1 protein expression and PARPi resistance [16] (S3 Fig). In Fig 4A, the resistance is clearly shown to all the PARPi when the SUM149B1.s* is grown with SUM149PT. Furthermore, the resistance is equivalent to the SUM149.A22 (Fig 4B). Similar is seen in SUM149 *53 BP1*$^{-/-}$ and SUM149 *SHLD1*$^{-/-}$ cells, where they have resistance to all four PARPi when compared to the parental cell line (Fig 4C and 4E). However, there is still some marginal sensitivity observed when compared to the SUM149.A22 cell line, where a 1.36 (p = 0.0004) and 1.72-fold (p = 0.04) change in area under the curve was observed in the SUM149.*SHLD1*$^{-/-}$ vs SUM149.22.*BRCA1*$^{+/-}$ competition assay when treated with veliparib and niraparib, respectively (S2B Fig and S2C Fig, Fig 4D and 4F). Further, in the SUM149.53 BP*1*$^{-/-}$ vs SUM149.A22.*BRCA1*$^{+/-}$ there was a significant 1.71 (p = 0.012), 1.42 (p = 0.003) and 1.52 (p = 0.009) AUC fold change in the olaparib, niraparib and talazoparib treatments, respectively, demonstrating that the resistant mutations do not create complete PARPi resistance when compared to a *BRCA1* reversion mutation in the context of these conditions.

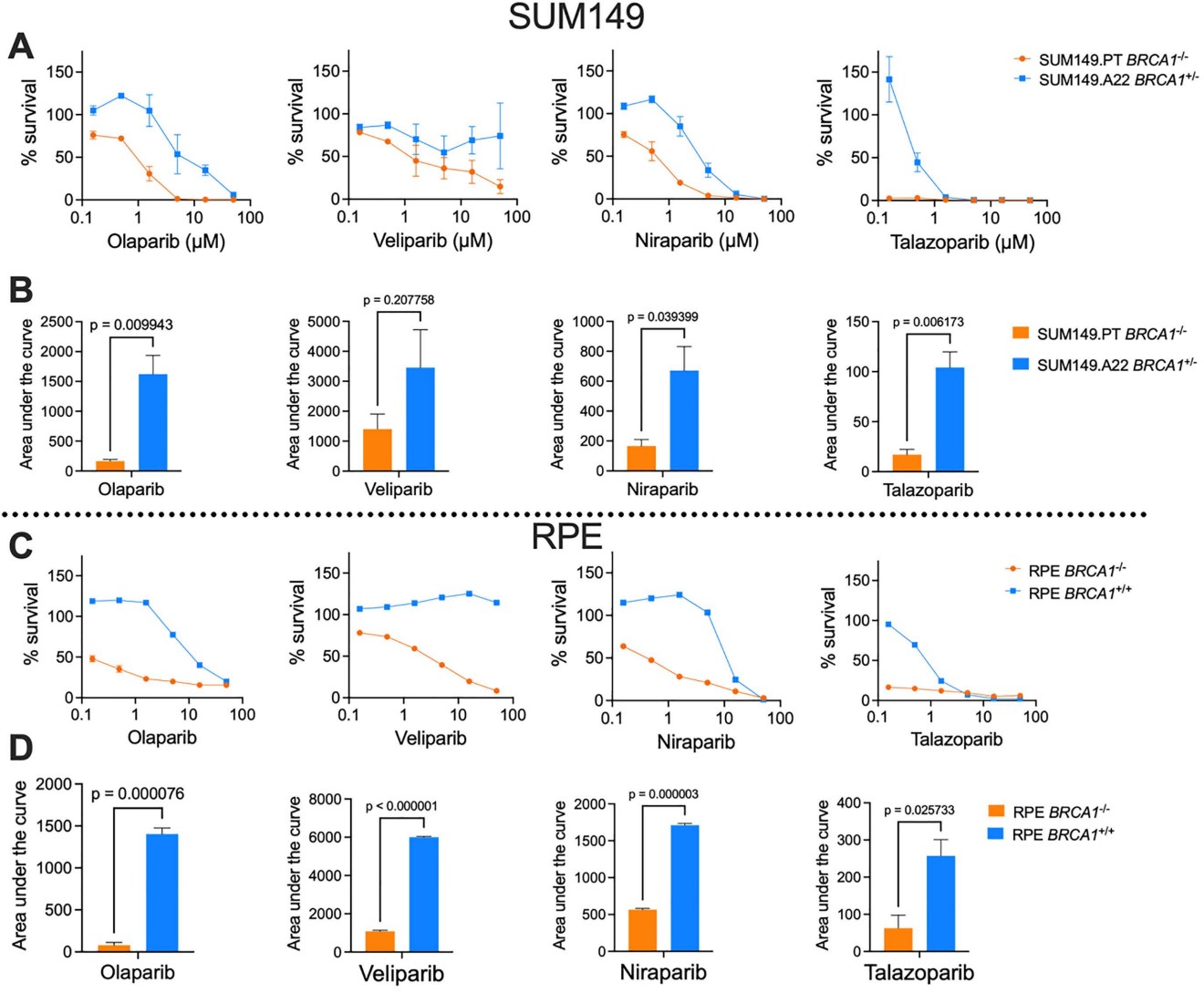

**Fig 3. Competitive growth assays performed with two isogenic *BRCA1* cell lines pairs, dosed with four different PARPi. A)** SUM149PT *BRCA1⁻/⁻* vs SUM149.A22 *BRCA1⁺/⁻* cell survival. **B)** Area under the curve comparison between the two competing cell lines in **A. C)** RPE *BRCA1⁻/⁻* vs RPE *BRCA1⁺/⁺* cell survival. **D)** AUC comparison between the two competing cell lines in **C.** Data is plotted as mean ± SEM where n = 3 for each treatment group. An unpaired t-test was performed comparing the AUC for both genotypes in B and **D.**

## Discussion

Synthetic lethal drugs have emerged as a transformative tool in precision oncology, offering targeted therapies that exploit genetic vulnerabilities in cancer cells while sparing healthy tissues [17]. The clinical success of PARPi in *BRCA1/2*-deficient cancers underscores the potential of synthetic lethal strategies to address previously hard-to-treat malignancies [18–20]. There is a need to expand the repertoire of synthetic lethal drug candidates to match and exploit the diversity of genetic alterations in cancers. To address this need, we developed and validated a competitive growth assay, designed to identify synthetic lethal interactions using isogenic cells lines grown in competition. While there is a substantial body of work that has identified genetic mechanisms of PARPi resistance, including genetic disruption of the Shieldin complex components and spontaneous *BRCA1* reversion mutations, less work has been published to isolate synthetic lethal

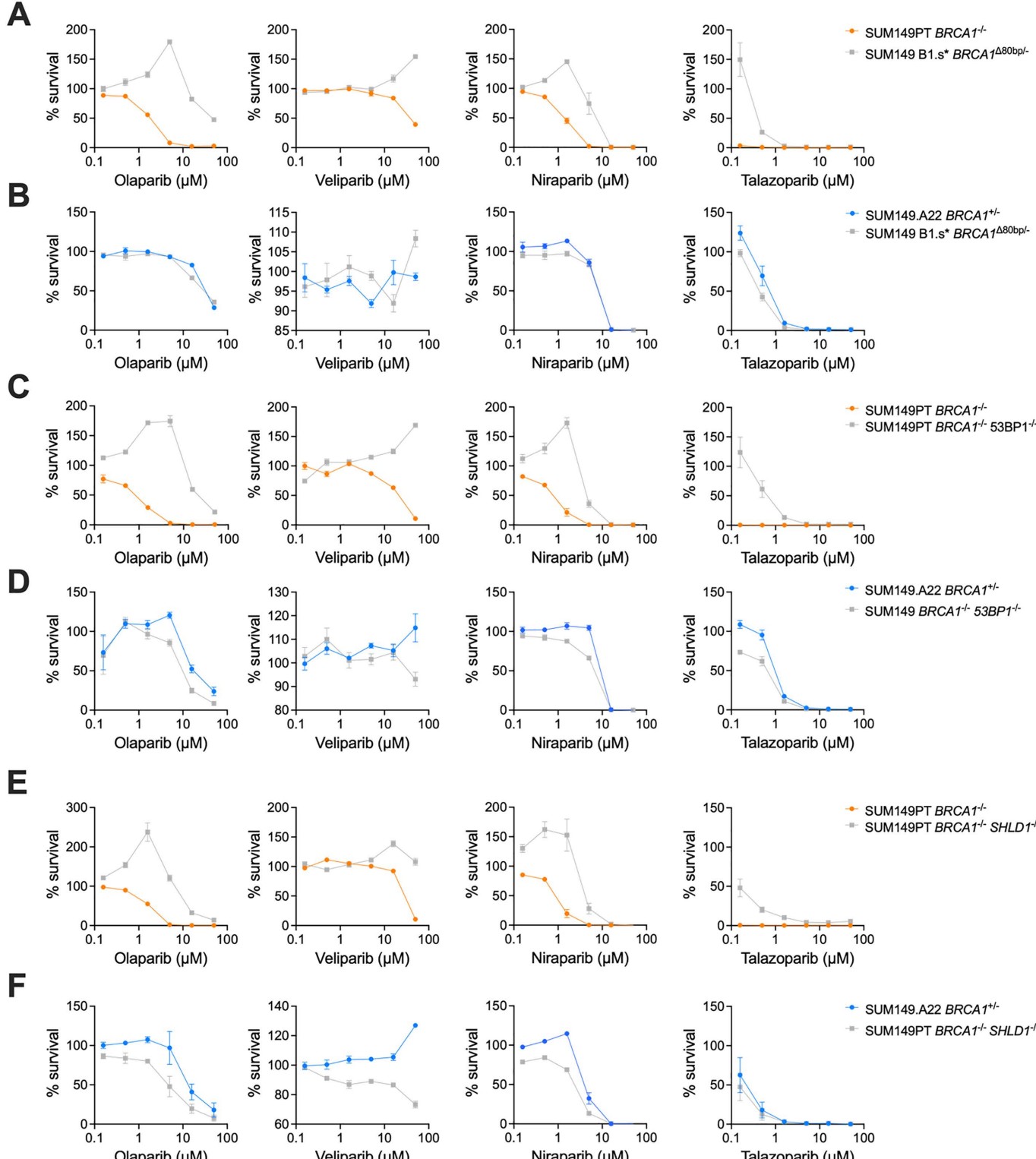

**Fig 4. Competitive growth assays with PARPi resistant SUM149 cell lines.** The four PARPi used were olaparib, niraparib, veliparib and talazoparib. Three SUM149 PARPi resistant cell lines were grown with either SUM149PT *BRCA1* $^{-/-}$ or SUM149.A22 *BRCA1* $^{+/-}$. The resistant cell lines were SUM149 B1.s* *BRCA1* $^{D80bp/-}$ (A) and **(B)**, SUM149PT *BRCA1* $^{-/-}$ *53 BP1* $^{-/-}$ (C) and (D) and SUM149PT *BRCA1* $^{-/-}$ *SHLD1* $^{-/-}$ (E) and **(F)**. The dosage range was 0.16 to 50 µM. Data is plotted as mean ± SEM where n = 3 for each treatment group.

targets using isogenic cell lines with stable genetic deletions or reversions with *BRCA1*, which is the context in which most cancers will need to be treated.

## Development of a new competitive growth assay to assess synthetic lethality

Here, our assay uses eGFP and mCherry to label each of the genotypes in isogenic cell line pairs from both cancer-derived and non-cancerous cell lines; SUM149PT and RPE, respectively. The assay has been performed as an endpoint assay where the cells are counted by flow cytometry on the final day. This makes the assay comparable to more common end-point cytotoxicity assays [21,22], for example the SRB assay. However, the assay could also be adapted to a live cell competitive growth assay to obtain further information about cell death or resistance dynamics for given drugs and geno-types. When investigating the synthetic lethal effects of olaparib with *BRCA1*-negative cells, the competitive growth assay performed similar to the SRB assay, where the fold change in AUC was not significantly different. Interestingly, the IC50's from the SRB assay were lower than what was calculated from the competitive growth assay. Further, cell survival dynamics appear to be different when grown in co-culture demonstrated by the differing olaparib IC50's between the competitive growth assay and the SRB assay. Additionally, IC50's of the same cell line and drug can differ between competitive growth assays depending on the other cell line it is competing with (S2 Table). This highlights how IC50's calculated from any survival assay are context-dependent, including SRB and competitive growth assays, and caution must be taken not to generalise results from a single method. For example, a competitive growth study using isogenic cell pairs to assess cis-platin resistance found that cisplatin-sensitive cells had a high rate of proliferation when cultured with its resistant isogenic counterpart. However, the higher proliferation rate increased sensitivity to cisplatin [23]. The increase in proliferation and sensitivity may be dependent on the drug- and cell-type. However, in this study, the SUM149 cell lines used had similar doubling rates, ranging from 26.5 h to 31.7 h (S3 Table) demonstrating the competitive advantages are due to the genes mutated and not the rate of cell proliferation.

In the competitive growth assay, *BRCA1*-proficient cells sometimes appear to have an increase in survival at low concentrations of PARPi. The spike in survival seems to occur at the dose in which the *BRCA1*-deficient cells start to have a decrease in survival and where there is the greatest survival difference between the two cells lines. The proliferation in the BRCA1-proficient cells could be caused by reduced competition for nutrients due to the decrease in the *BRCA1*-deficient cells. Irrespective of the cause of the differential genotype-dependent response, it highlights the utility of this assay as a complement to survival assays that use a single genotype per well.

## Assay validation

Three of the four PARPi used to validate the competitive growth assay are FDA approved for the treatment of *BRCA1/2* negative breast and/or ovarian cancers (olaparib, niraparib and talazoparib) [18–20]. In this assay, olaparib, niraparib and veliparib all demonstrated similar potency with greater than 50% survival observed for *BRCA1* negative cells at the lowest dose of 158 nM. However, talazoparib was an outlier with its potency, inducing near-complete cell death in *BRCA1* negative cells at the same 158 nM dose. This is reflected in the clinical dosage guideline for these PARPi inhibitors where talazoparib dosed at 1 mg per day, compared to niraparib and olaparib, which are dosed at 300 mg per day and 300 mg twice per day, respectively [24,25].

## PARPi resistance mechanism analysis

The competitive growth assay was also able to replicate *BRCA1* negative cell-PARP inhibitor resistance results, that had previously been observed in *53 BP1* KO, *SHLD1* KO and *BRCA1* reversion mutations. *BRCA1* reversion mutations have shown complete resistance to PARPi *in vitro* [16]. However, only partial, yet significant, resistance has been in observed with the *53 BP1* and *SHLD1 KO* cells [14,26]. In our study, we observed complete resistance in the SUM149 *BRCA1⁻/⁻ 53*

BP1⁻/⁻ to veliparib and talazoparib treatments (S2 Fig), and significant but partial resistance in the SUM149 *BRCA1⁻/⁻ SHLD1⁻/⁻* cells when treated with olaparib, veliparib and niraparib, reflecting results seen in the previous studies.

### Future directions

The similar dose-response curves to each drug across the two *BRCA1* isogenic cell lines used in the study highlights the robustness of the *PARP-BRCA1* synthetic lethality. A key advantage of this assay is its potential to be upscaled to screen new synthetic lethal drugs especially when there is a synthetic lethal interaction that is as clear as the *PARP-BRCA1* interaction. *PARP-BRCA1* synthetic lethality relies on the function of adjacent DNA repair pathways, demonstrating that synthetic lethality is context-dependant and functions in a system that contains more than just two target genes [27]. This competitive growth assay could be expanded beyond two fluorescently marked cell types and be used to help characterise more complex synthetic lethal relationships. The number of genes and cell lines that could be used in competition is only limited by the number of fluorescent reporters that can be transformed into the cells and detected by flow cytometry.

Growing different cell types in competition has been used numerous times before to address various research questions [10,11,14,15]. One potential improvement to this competitive growth assay would be to perform the assay longitudinally by counting the cells using microscopy over regular intervals. This approach has previously been performed when validating hits from a CRISPR-based PARPi resistance screen [14]. An isogenic cell line was made with the PARPi resistance gene knocked out in one of the cell line pair. Each genotype was fluorescently labelled, and the cells were observed via microscopy periodically over 18 days. This data can show the exact time point when the resistant cells start to have a competitive advantage. Furthermore, the competitive growth assay also helps replicate, in isolation, the conditions of a CRISPR screen, where all knockout clones are grown together in the same dish.

A competitive growth assay could also be applied in 3D cell cultures using fluorescent reporters to track the growth of different cell types [11]. While 3D cultures are less high throughput than standard 2D culture, modelling competitive growth in 3D could provide a more accurate representation of *in vivo* cell survival and enhances *in vitro* pre-animal studies of new drug treatments. This assay is not limited to *BRCA1* synthetic lethal relationships and could potentially be applied to any cell or tumour type that has a genetic vulnerability that leads to a selective pharmaceutical sensitivity in 2D or 3D.

### Study limitations

The competitive co-culture assays developed and validated here do not always have a wide therapeutic window than the non-competitive SRB cytotoxicity assay, however the data it provides can improve our understanding of how synthetic lethality occurs *in situ* and can be used to study synthetic lethality temporal dynamics. The dose-response curves often show a dose where the there is a survival increase in the *BRCA1*-positive cells, for example, in Fig 2A. This allows us to find a dose in which the maximum toxicity is observed in the *BRCA1* negative cells, while giving the *BRCA1* positive cells the greatest competitive advantage. Such dose curves are not observed in the non-competitive SRB assay. Although, the biological relevance of the competitive growth assay is still limited by 2D cell culture, in which it is important to note that the assay lacks the diverse array of cell types and structures that would be present around the site of a tumour.

### Supporting information

**S1 Fig. The sequence of the *BRCA1* mutation in SUM149PT that was corrected to make isogenic cell pair.** In red is where the c.2169delT mutation is and subsequent correction mutation. In green are silent mutations which create new BseYI and AvaI restriction sites that can be used for genotyping the SUM149.A22 correction.
(TIFF)

**S2 Fig. Area under the curve comparison bar graphs of PARPi resistance genes treated with PARPi.** (A) – SUM149.A22 vs SUM149 B1.s*, (B) SUM149.A22 vs SUM149 *53 BP1* and (C) SUM149.A22 vs SUM149 *SHLD1*). These

plots are derived from the data in Fig 4, in which they show statistical significance between the survival curves of the cell lines in the competitive growth assay. Unpaired t-tests were performed between each isogenic cell pair in each treatment. (TIFF)

**S3 Fig.** *BRCA1* **expression in RPE and SUM149 cell lines visualised using western blot.** Western blot was performed on 40 μg of cell lysate from RPE *BRCA1*$^{+/+}$, RPE *BRCA1*$^{-/-}$, SUM149PT, SUM149.A22 and SUM149 B1.s*. The membranes were stained with Revert™ 700 total protein stain as a loading control.
(PDF)

**S1 Table. Background information on cell lines used in the competitive growth assays.**
(DOCX)

**S2 Table. Summary of IC50's and area under the curve figures for each of the competitive growth assays.**
(DOCX)

**S3 Table. SUM149 cell line doubling times.**
(DOCX)

**S1 Data. Normalized data.**
(XLSX)

**S2 Data. Raw western images.**
(PPTX)

## Acknowledgments

The SUM149 cell line series (SUM149 parental, SUM149 B1.S*, SUM149 TP53 BP1, SUM149 SHLD1 (C20orf196) were shared by Stephen Pettitt (The Institute of Cancer Research, UK). The *BRCA1*-edited SUM149.A22 cell line was generated by the Monash Genome Modification Platform (MGMP). The authors acknowledge the facilities, and the scientific and technical assistance of MGMP, Monash University. MGMP is supported by Therapeutic Innovation Australia (TIA). TIA is supported by the Australian Government through the National Collaborative Research Infrastructure Strategy (NCRIS) program.

## Author contributions

**Conceptualization:** Michael F Sharp, Wayne Crismani.

**Formal analysis:** Shiella A Soetomo, Michael F Sharp.

**Funding acquisition:** Shiella A Soetomo, Wayne Crismani.

**Methodology:** Shiella A Soetomo, Michael F Sharp.

**Supervision:** Michael F Sharp, Wayne Crismani.

**Writing – original draft:** Shiella A Soetomo, Michael F Sharp.

**Writing – review & editing:** Shiella A Soetomo, Michael F Sharp, Wayne Crismani.

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
