## [Decision Letter · Decision Letter 0]

11 Jun 2025

Dear Dr. Crismani,

Thank you for submitting your manuscript to PLOS ONE. After careful consideration, we feel that it has merit but does not fully meet PLOS ONE’s publication criteria as it currently stands. Therefore, we invite you to submit a revised version of the manuscript that addresses the points raised during the review process.

We look forward to receiving your revised manuscript.

Kind regards,

Zu Ye, Ph.D.

Academic Editor

PLOS ONE

Journal Requirements:

2. Please update your submission to use the PLOS LaTeX template. The template and more information on our requirements for LaTeX submissions can be found at http://journals.plos.org/plosone/s/latex .

“W.C. receives a fellowship related to this work from the  Victorian Cancer Agency (MCRF21006). SVI receives Operational Infrastructure Support from the Victorian State Government. S.S. received an Indonesian Lembaga Pengelola Dana Pendidikan scholarship.”

4. In the online submission form, you indicated that “We are very happy to make the cell survival data from our paper available in raw format, or pre-processed. Please feel free to contact me to explain for this sort of article what you would like shared and we can upload it as a supplementary table.”

5. We notice that your supplementary figures are included in the manuscript file. Please remove them and upload them with the file type 'Supporting Information'. Please ensure that each Supporting Information file has a legend listed in the manuscript after the references list.

Reviewers' comments:

Reviewer's Responses to Questions

**Comments to the Author**

1. Is the manuscript technically sound, and do the data support the conclusions?

Reviewer #1: Yes

Reviewer #2: Partly

Reviewer #3: Yes

Reviewer #4: Yes

2. Has the statistical analysis been performed appropriately and rigorously?

Reviewer #1: Yes

Reviewer #2: No

Reviewer #3: Yes

Reviewer #4: I Don't Know

3. Have the authors made all data underlying the findings in their manuscript fully available?

Reviewer #1: Yes

Reviewer #2: No

Reviewer #3: Yes

Reviewer #4: Yes

4. Is the manuscript presented in an intelligible fashion and written in standard English?

Reviewer #1: Yes

Reviewer #2: Yes

Reviewer #3: Yes

Reviewer #4: No

Reviewer #1: This manuscript presents a competitive two-color co-culture assay designed to assess synthetic lethality and PARP inhibitor responses in BRCA1-deficient and resistant cell models. The system is well validated using both cancerous and non-cancerous isogenic cell lines, and the inclusion of resistance models (e.g., BRCA1 reversion, 53BP1-/-, SHLD1-/-) adds significant value to the findings.

The methods are clearly described and technically sound. The comparison to standard SRB assays provides useful context and highlights differences in IC50 values and dynamic responses between assay types. Flow cytometry-based quantification is appropriately used. The conclusions are well supported by the data.

A few minor revisions are recommended:

- Clarify the cause or implication of the discrepancy in IC50 values observed between the co-culture and SRB assays.

- Discuss the observed growth advantage of BRCA1-proficient cells at low PARPi doses, and whether this has biological or technical relevance.

- Consider expanding briefly on the assay’s potential adaptability to additional tumor types or to 3D culture models, as this could broaden impact.

The manuscript is clearly written and meets standards for English usage. Data availability and ethical compliance are appropriately addressed. No concerns were identified regarding dual publication, ethics, or research integrity. The study does not involve dual use research of concern.

Recommendation: Accept with minor revisions.

Reviewer #2: This manuscript lacks significant advancements in the described protocol and omits essential information regarding experimental sources.

While the authors have focused on the methodology, they have failed to provide sufficient details about the cell lines used. To complete this project, the authors must clarify the genetic and phenotypic characteristics of the cells or, at the very least, provide full references for both wild-type and mutant cell lines. Due to these omissions, many sections of the manuscript may lead to misleading conclusions.

Additionally, the manuscript's structure is confusing; the authors should separate the main text from the figures and figure legends. In conclusion, substantial revisions are needed—both in terms of additional data and structural organization—before this manuscript can be considered for publication.

Reviewer #3: The study design is well executed with the data figure well assembled and easy to be understood. The statistical analysis is also clear. The manuscript by virtue of its results is a solid candidate for publication if the authors can address my comments included in the attached review report.

Reviewer #4: 2) For this comment, the n numbers is 8 in fig 2, 3 in fig 4 and have not been mentioned in fig 3. Perhaps the n numbers should have been consistent.

4) I have added in corrections for grammatical errors as well as structure of sentences used in the attached file. There are a few errors with the tenses as well which I have also pointed out.

**Do you want your identity to be public for this peer review?** For information about this choice, including consent withdrawal, please see our Privacy Policy

Reviewer #1: **Yes: ** Chu Kwen Ho

Reviewer #2: No

Reviewer #3: No

Reviewer #4: **Yes: ** Aleena Khalid

---

## [Author Response · Author response to Decision Letter 1]

19 Aug 2025

Please see attached document for a complete point-by-point response to the Reviewers.

---

## [Decision Letter · Decision Letter 1]

5 Sep 2025

Differential PARP inhibitor responses in BRCA1-deficient and resistant cells in competitive co-culture

PONE-D-25-26053R1

Dear Dr. Wayne Crismani,

We’re pleased to inform you that your manuscript has been judged scientifically suitable for publication and will be formally accepted for publication once it meets all outstanding technical requirements.

Kind regards,

Zu Ye, Ph.D.

Academic Editor

PLOS ONE

Additional Editor Comments (optional):

Reviewer #1:

Reviewer #3:

Reviewers' comments:

Reviewer's Responses to Questions

**Comments to the Author**

Reviewer #1: All comments have been addressed

Reviewer #3: All comments have been addressed

2. Is the manuscript technically sound, and do the data support the conclusions?

Reviewer #1: Yes

Reviewer #3: Yes

3. Has the statistical analysis been performed appropriately and rigorously?

Reviewer #1: Yes

Reviewer #3: Yes

4. Have the authors made all data underlying the findings in their manuscript fully available?

Reviewer #1: Yes

Reviewer #3: Yes

5. Is the manuscript presented in an intelligible fashion and written in standard English?

Reviewer #1: Yes

Reviewer #3: Yes

Reviewer #1: The authors have carefully addressed all of my prior concerns. Specifically, they now provide a clear explanation for the discrepancy in IC50 values observed between the SRB and competitive co-culture assays, emphasizing the context-dependence of survival assays. They have also thoughtfully discussed the apparent growth advantage of BRCA1-proficient cells at low PARPi concentrations, offering both a rationale and a statement on the technical relevance of this observation. Finally, the manuscript has been strengthened by the expanded discussion of the assay’s potential adaptability to additional tumor types and to 3D culture models, which enhances its broader applicability and impact.

Overall, the revisions are thorough, well-reasoned, and improve the clarity and significance of the work. I have no further concerns, and I recommend the manuscript for publication in its current form.

Reviewer #3: All of my comments/suggestions have been well addressed by the authors.The revised manuscript is a good candidate for publication.

**Do you want your identity to be public for this peer review?** For information about this choice, including consent withdrawal, please see our Privacy Policy

Reviewer #1: No

Reviewer #3: No
